# Improving understanding of groundwater flow in an alpine karst system by reconstructing its geologic history using conduit network model ensembles

Chloé Fandel[1], Ty Ferré[2], François Miville[3], Philippe Renard[3], Nico Goldscheider[4]

[1]Geology Department, Carleton College, Northfield, MN 55057, USA
[2]Department of Hydrology & Atmospheric Sciences, University of Arizona, Tucson, AZ, 85721, USA
[3]Centre d'Hydrogéologie et de Géothermie, Université de Neuchâtel, Neuchâtel, 2000, Switzerland
[4]Institut für Angewandte Geowissenschaften, Karlsruher Institut für Technologie, Karlsruhe, 76131, Germany

*Correspondence to*: Nico Goldscheider (nico.goldscheider@kit.edu)

**Abstract.** Reconstructing the geologic history of a karst area can advance understanding of the system's present-day hydrogeologic functioning, and help predict the location of unexplored conduits. This study tests competing hypotheses describing past conditions controlling cave formation in an alpine karst catchment, by comparing an ensemble of modelled networks to the observed network map. The catchment, the Gottesacker karst system (Germany/Austria), is drained by three major springs and a paleo-spring, and includes the partially explored Hölloch cave, which consists of an active section whose formation is well-understood, and an inactive section whose formation is the subject of debate. Two hypotheses for the
formation of the inactive section are: 1) glaciation obscured the three present-day springs, leaving only the paleo-spring, or 2) the lowest of the three major springs (Sägebach) is comparatively young, so its subcatchment previously drained to the paleo-spring. These hypotheses were tested using the pyKasso Python library (built on anisotropic fast marching methods) to generate two ensembles of networks, one representing each scenario. Each ensemble was then compared to the known cave map. The
simulated networks generated under Hypothesis 2 match the observed cave map more closely than those generated under Hypothesis 1. This supports the conclusion that the Sägebach spring is young, and suggests that the cave likely continues southwards. Finally, this study extends the applicability of model ensemble methods from situations where the geologic setting is known but the network is unknown, to situations where the network is known but the geologic evolution is not.

## 1 Introduction

Mapping subsurface cave and conduit networks in karst systems provides crucial information for water resource management, hazard prevention, archaeological research, and the protection of important ecological and cultural sites. Even small-diameter conduits play an important role in groundwater flow. However, mapping conduit networks is a challenging task (Trimmis, 2018, Sellers and Chamberlain, 1998). Conduits can extend far below the surface, and speleologists attempting to explore them may experience dangers such as flash floods, falling rocks, hypothermia, and disorientation. Even in well-explored systems,

the full network is impossible to map, because small-diameter conduits (<0.5 m) are inaccessible to humans and difficult to detect by geophysical methods (Jaquet and Jeannin, 1994).

Understanding the speleogenetic processes driving cave formation and the geologic history of an area can help guide cave exploration, data collection, and groundwater resource management. Karst systems commonly evolve over time in several phases, during which different climate and geologic conditions control which parts of the system (e.g. diffuse recharge zones,

focused inlets, outlets, conduits) are actively transmitting flow. Conduits or springs that see active flow during one phase of karstification may go dry when conditions change. Although these older parts of the system may become inactive, they are still open and can be reactivated during extreme flow events or when climate conditions shift again (Audra and Palmer, 2011). An experienced geologist may be able to reconstruct how the karst system functioned in the past, and therefore predict the likely locations of presently inactive conduits. These predictions can sometimes be tested through targeted exploration of the

locations where caves are suspected to exist. Some hypothesized conduits, however, may be inaccessible, making it impossible to confirm or refute their existence through exploration. In other cases, several different historical scenarios may appear equally plausible, making it difficult to determine how the system evolved. As a result, geologic intuition alone may not be sufficient to fully understand a karst area. This can lead to knowledge gaps or misconceptions that limit the reliability of future simulations of the system's behavior under projected climate or land use scenarios that differ from present-day conditions.

**2 Approach**

This study uses a model-based approach to identifying the geologic processes which could explain the formation of a particular cave in the Gottesacker karst system. In this system, detailed cave maps are available in parts of the conduit network.

While some portions of the explored cave system are very active, containing an accessible cave stream with open-channel flow that connects to a major karst spring, other portions of the explored system are now inactive, and are thought to have formed

when past conditions were different from the present configuration of the system. However, two different explanations are possible: 1) glaciation obscured all of the present-day springs, leaving only the paleo-spring as the primary drainage point for the entire karst system, or 2) the lowest of the major present-day springs is significantly younger than the others, so the paleo-spring and the two uppermost springs jointly drained the aquifer system.

These competing hypotheses were tested using a computationally efficient karst network model to generate probability maps

of possible conduit locations under each of the proposed past scenarios. The simulated conduit networks under each scenario were then compared with the mapped portion of the inactive cave system, to determine which scenario best matched the observed network. Finally, the analysis was extended to propose possible locations of the unmapped parts of the system.

## 3 Field site: Gottesacker

The modeling approach demonstrated in this study was tested on the complex, extensively studied Gottesacker karst system
in the German/Austrian Alps, described in Goldscheider (2005). This 35 km$^2$ catchment consists of a series of plunging
synclines and anticlines draining to the Schwarzwasser valley, which cuts roughly perpendicularly across the fold axes (Figure
1). The karst aquifer lies north and northwest of the valley in a limestone unit widely exposed at the surface. It is locally
overlain by sandstone and younger units, and underlain by non-karstifiable marl and older units.

This karst system is also the site of one of the longest caves in Germany, the 12 km Hölloch cave in the Mahdtal valley, which
has been the subject of avid exploration and of several books and documentary films since the early 1900s (Höhlenverein
Sonthofen, 2006). The local caving club has created and shared a detailed map of the explored portion of the cave. The northern
part of the cave (trending SE) is active and contains a cave stream, while the southern part (trending NNW) is inactive and
does not have any continuous underground drainage or cave streams.

Three major springs drain the system: an estavelle (QE, elevation 1120 m a.s.l.), the Aubach spring (QA, elevation 1080 m
a.s.l.), and the Sägebach spring (QS, elevation 1035 m a.s.l.). The estavelle acts as swallow hole during low-flow conditions
but acts as a spring during high-flow conditions. A paleo-spring (QO, elevation 1190 m a.s.l.) partway up the Mahdtal valley,
near Höflealpe, remains dry for timespans of years to decades. In 2005, QO temporarily re-activated, during extreme high flow
conditions (Figure 2). This is the only recorded flow event ever observed at this spring.

QO is located at a geological contact between the karstified limestone aquifer and overlying low-permeability formations
(sandstone and marl). Upstream from QO, the karstified limestone is exposed at the land surface, so precipitation infiltrates
rapidly and does not generate any significant surface flow. However, since QO is located directly above and is connected to
the cave stream in the conduit system beneath it, any surface water that does exceed the infiltration capacity of the karst or is
injected at QO would reach the cave stream and flow to Sägebach spring (QS). Downstream of QO, the overlying sandstone
and marl in the core of the syncline confine the karst aquifer, which drains to the Sägebach spring (QS). QO can therefore act
as an overflow when the discharge capacity of QS is exceeded, and large-scale backflooding occurs, as observed in 2005. In
this lower part of the area, extending into the Schwarzwasser valley, the karst aquifer was (and partly still is) confined by
overlying low-permeability sandstone and marl and by widespread moraine deposits.

South of the main Schwarzwasser valley, non-karstifiable flysch and marl lithology prevents conduit development, and
drainage occurs instead through a network or surface streams. Several other geologic units and small springs are present, but
for the purposes of conduit modeling, the geology was represented using a simplified model focused on delineating the
boundaries of the karstifiable limestone unit.

The general configuration of the conduit network (referred to in the rest of this paper as the *expected network*) was originally
defined by Chen and Goldscheider (2014), and refined by Chen et al. (2018), based on geologic mapping by Wagner (1950),

predominant fracture orientations documented by Cramer (1959), several decades of speleological investigations by the regional caving club (Höhlenverein Sonthofen), 18 quantitative multi-tracer tests by Goldscheider (2005), Göppert and Goldscheider (2008), and Sinreich et al. (2002), and hydrogeological field observations by Goldscheider (2005). Anticlinal

95  axes act as groundwater divides sectioning the region into sub-basins, each drained by a major conduit following synclinal axes. These conduits merge into a primary conduit that parallels the surface stream in the Schwarzwasser valley. The expected network map does not represent exact conduit locations, but rather the general configuration and approximate location of the conduits. This network map has been used successfully for several previous groundwater flow modeling efforts (Chen and Goldscheider, 2014, Chen et al., 2017, Chen et al., 2018, Fandel et al., 2021).

100

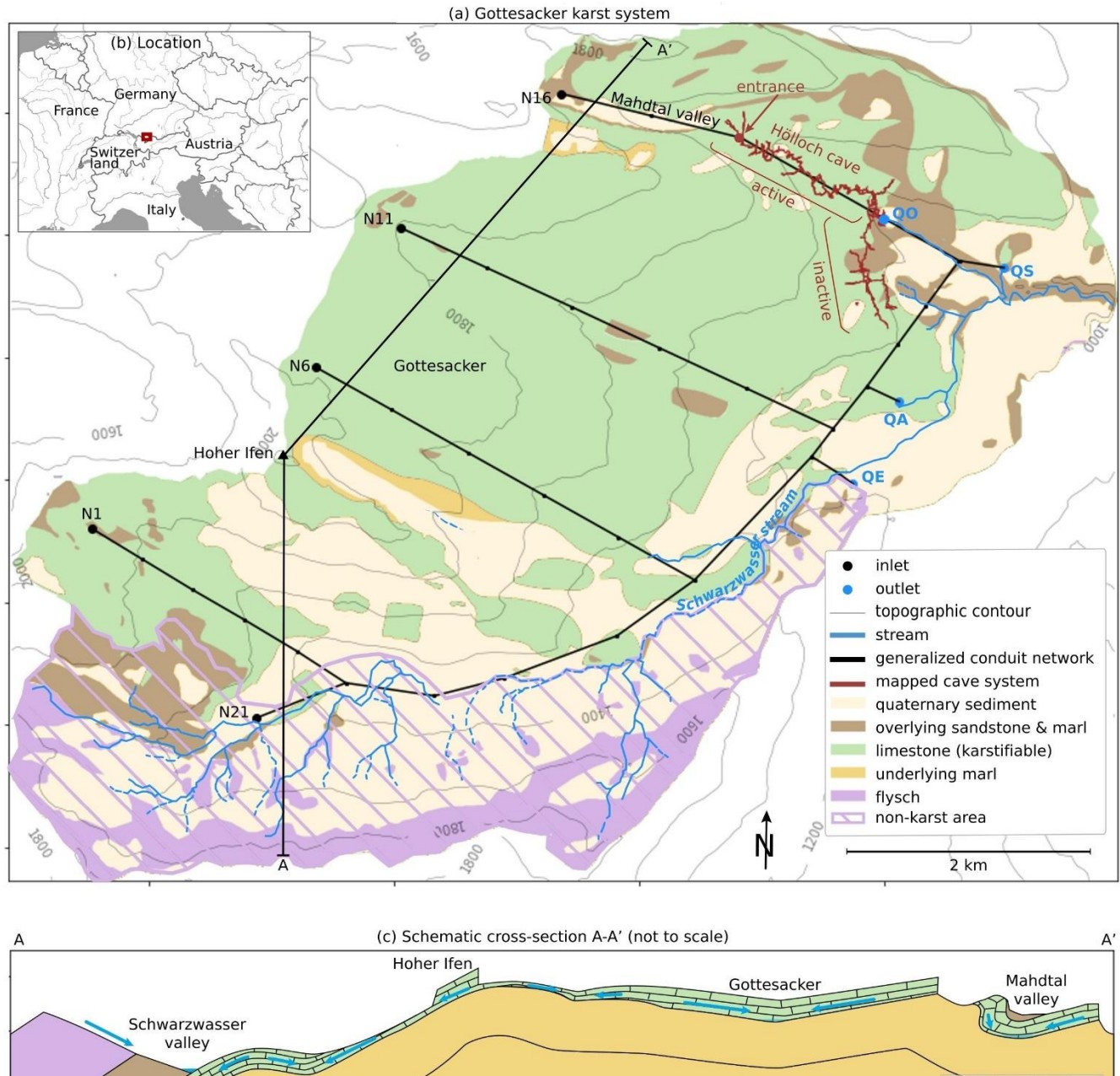

**Figure 1: Overview of the Gottesacker karst system. (a) Simplified geologic map after Fandel et al. (2021). The karst aquifer is located in the limestone unit, and drains to the three springs in the lower part of the system (QE, QA, QS). The paleo-spring (QO) is only active during extreme high-flow conditions. All springs flow into the Schwarzwasser stream, shown in blue. Several small tributaries also feed the stream from the south. Outlets correspond to mapped karst springs after Goldscheider (2005). Expected conduit network and inlet names after Chen et al. (2018) (b) Location within Europe. Basemap: ESRI. (c) Schematic cross-section along line A-A', without showing Quaternary deposits, adapted from Goldscheider (2005).**

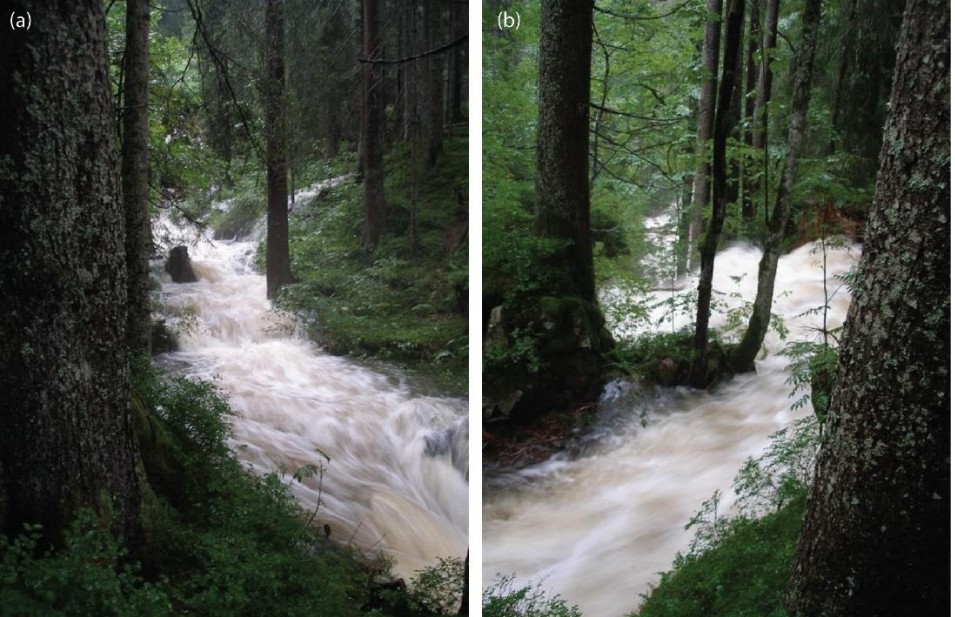

**Figure 2: Two views of the paleo-spring (QO) re-activating during extreme high-flow conditions in 2005. This is the only observed flow event at this spring, which is completely dry most of the time. Photos by K. Kessler.**

### 4 Hypotheses

Two different hypotheses are currently under consideration to explain the formation of the inactive part of the Hölloch cave system:

*Hypothesis 1:* The Schwarzwasser valley was covered by a glacier, overlying the karstifiable units up to the elevation of the paleo-spring (QO), which was the primary spring draining the entire system.

*Geologic evidence:* Widespread moraines and other geomorphologic evidence indicate the presence and extent of the main glacier in the Schwarzwasser valley, which covered all of locations of present-day valley springs (QE, QA, QS) and filled the valley up to an elevation just below the paleo-spring (QO), located approximately 200 m above the valley floor. The glacier appears to have filled the rest of the valley with roughly the same thickness of ice, creating a boundary just below and parallel to the inactive portion of the mapped conduits. There is clear geomorphologic evidence that the entrance shaft of the Hölloch cave was formed by sinking glacial meltwater, and was an active inlet during and after glaciation. The streambed created by meltwater flowing down the valley is still visible, and still channels temporary surface flows during snowmelt or intense rainfall. End moraines further up the Mahdtal valley indicate the presence of a local glacier in the upper part of this valley, as well as in the upper part of the valley located in the breached syncline northeast of Hoher Ifen.

*Hypothesis 2:* Low-permeability sedimentary formations covered the Sägebach spring (QS), so the paleo-spring (QO) drained the Mahdtal valley and connected to the Aubach spring (QA), which was at the time the lowest spring in the system.

*Geologic evidence:* Field evidence indicates that the Sägebach spring (QS) is much younger than the Aubach spring (QA), but it is difficult to estimate the exact age difference. QA is located in a large gorge (hundreds of meters long and tens of meters deep), with many open fractures and cavities and a cave entrance. It is in the middle of a zone of exposed and highly karstified limestone. It does not appear to have an upper limit of discharge, and looks like a very well-developed, older outlet for the karst aquifer system. QS, on the other hand, is located along a fault, in a short, narrow, and shallow gorge. The karstified limestone is only very locally exposed and is surrounded and overlain by younger, less permeable units (sandstone, marl, and glacial moraine deposits). QS has limited discharge, causing back-flooding when the discharge capacity of the spring is exceeded. This geologic setting suggests very young (probably post-glacial) exposure by erosion of the confining layers along the fault.

Both of these scenarios are supported by field observations, so the question is not whether they occurred, but whether they were determinant for the formation of the now-inactive portion of the cave network. This depends both on the temporal order and on the duration of the different processes at work. One of the reasons for this question is that we do not have enough clear field evidence to determine whether the glaciation preceded or followed the uncovering of QS.

## 5 Method: Anisotropic fast marching to generate probability maps

To test these hypotheses, many possible network configurations were modelled using a stochastic simulation method implemented in the Python karst modelling package pyKasso (Fandel et al., 2022). This package implements the SKS approach originally proposed by Borghi et al. (2012), but uses an anisotropic fast marching algorithm (Sethian, 1999, Mirebeau, 2014). Anisotropic fast marching algorithms calculate the optimal path from one point to another through a medium, in which the ease of travel varies both spatially and directionally. Karst conduits can be simulated using this type of algorithm based on the assumption that a conduit represents the fastest path from an inlet (such as a doline or swallow hole) to an outlet (a spring) (Borghi et al. 2012). Luo et al. (2021) also implemented the use of anisotropic fast marching to simulate karst systems, with a focus on representing the anisotropy resulting from a fracture network by using 3D fields of equivalent permeability tensors. The SKS approach presented by Borghi et al. (2012) is three-dimensional, but isotropic. The approach presented by Luo et al. (2021) is 3D and anisotropic. As in SKS and in pyKasso, this last approach requires generating a discrete fracture network model, but then it estimates by iterative calculations the size of the representative elementary volume, and computes the equivalent permeability tensors for all the cells of the model by solving multiple local flow problems in different directions. The anisotropic fast marching is then carried out using the upscaled field of spatially varying equivalent permeability tensors. The advantage of this approach is that it permits the representation of fractures at a subgrid scale. The preprocessing calculations are however rather computationally intensive. The pyKasso package presented in Fandel et al. (2022) is 2D (with

a more recent 3D version now functioning), and uses anisotropic fast marching in a slightly different framework. It focuses on
enabling rapid generation of numerous possible conduit networks influenced by multiple factors, such as the topography of
the geologic surface on which karst conduits develop. To test the hypotheses presented in this paper, the 2D anisotropic version
of pyKasso was chosen, because of its computational efficiency in generating numerous possible conduits, and its ability to
consider the effects of bedding plane topography, spatial variations in travel cost, and inlet/outlet pairings, in addition to
fracture networks. For karst systems in which the conduits are more strongly influenced by the facture network than by the
structure of the contacts between geologic units, and in which conduits develop equally extensively in all three dimensions,
rather than mostly in the x and y directions, Luo et al.'s implementation may be a better choice.

In pyKasso, the travel medium represents the geologic setting, in which some rock units are more soluble than others (i.e.
easier for conduits to travel through). Conduits are also assumed to form preferentially in certain orientations: in the direction
of the maximum downward hydraulic gradient, and/or along the dip direction of bedding planes (Audra and Palmer, 2015;
Dreybrodt et al., 2005; Palmer, 1991). The user can specify which plane should influence conduit orientation. For shallow,
unsaturated karst systems with an impermeable basal unit such as the example presented here, the gradient of the lower contact
between the karstifiable unit and the underlying non-karstifiable unit is the dominant influence, because water recharging
downwards from the land surface encounters a low-permeability barrier and subsequently flows laterally along this low-
permeability surface (Filipponi, 2009).

Stochasticity can be introduced to the simulations through the generation of a unique fracture network for each model
realization, based on the statistical distribution of fracture families observed in the field. The statistical metrics describing the
fracture network are the density, the minimum and maximum strike, and the minimum and maximum length. For a table of
fracture statistics, see Fandel et. al. (2022). Fractures are assumed to be easier for conduits to travel through than the
surrounding rock.

Additional controls on the configuration of the simulated network are possible by iterating over multiple phases of karst
development, and over multiple outlets. The inlets to the system can also be divided into groups and assigned to separate
outlets, representing different subcatchments within the larger system. These aspects tend to contribute more variability to the
resulting networks than stochasticity arising from fractures only. All these aspects can be controlled in a deterministic or
stochastic manner depending on the available information for a given site (see Fandel et al., 2022 for a full explanation of
possible variations).

The primary advantages of fast marching methods are their low data requirements and their computational efficiency compared
to other conduit network models. These characteristics allow the rapid simulation of hundreds of network realizations for a
single site (for this study, 100 simulations ran in under two minutes on a laptop with a 2.7 GHz dual-core i7 processor and 16
GB RAM). However, this approach does not represent the actual physical and chemical processes driving speleogenesis.
Unlike more complex speleogenetic models, each individual pyKasso model realization, because it is based on very little
information, is unlikely to capture the true network configuration. Instead, this method is more appropriate for rapidly

exploring various scenarios and generating probability maps: visual representations of a model ensemble indicating the likelihood of a simulated conduit being present at any given location.

## 5.1 Quality check: accuracy of network simulations

To represent the Gottesacker karst system with pyKasso, the medium through which conduits form was bounded using a three-dimensional geologic model, created using the Python package GemPy (de la Varga et al., 2019). For a detailed description of the geologic model, see Fandel et al. (2021). For use with pyKasso (only available in 2D at the time of this work), the 3D extent of the karstifiable limestone unit was projected onto a 2D plane, and used as the initial cost map (see Table 1 for travel cost values). The resulting 2D geologic map has a resolution of 181 x 141 cells, with 50 m x 50 m pixels (Figure 3b & c). This

resolution was chosen because it is detailed enough to represent the criteria used to test the hypotheses under consideration in this study (the general location and orientation of conduits), but coarse enough to maintain the fast computation times needed to run hundreds of simulations. A model with 5 m x 5 m pixels was originally tested, but was discarded because it significantly increased the computation time (nearly 50 minutes compared to less than 2 minutes to run 100 simulations) without significantly increasing the degree of confidence with which the study questions could be answered. The gradient driving the

preferred orientation of conduit formation was calculated from the surface of the contact between the base of the karstifiable limestone and the less-permeable underlying marl. Previous work by Filipponi (2009) demonstrated that in the stratigraphic sequence present at this study site, where the Schrattenkalk limestone is the primary karstifiable unit, conduits form preferentially along the lower contact between the Schrattenkalk and the underlying Drusberg marl, parallel to bedding planes. Outlet coordinates were assigned based on mapped spring locations (Goldscheider, 2005). Inlet coordinates and inlet/outlet

pairings were assigned based on locations and springshed boundaries inferred from tracer test evidence and geologic structure in previous flow modeling efforts (Chen and Goldscheider, 2014). For a full description of the original conduit model parameters, results, and limitations, see Fandel et al. (2022). The conduit model presented in this paper is modified from that in Fandel et al. (2022) in that it uses as input the 2D actual areal extent of the karstifiable unit (rather than a map of where the limestone crops out at the land surface) for the cost maps, and the altitude map of the contact surface between limestone and

marl (rather than the land surface) for the gradient calculations. The inlet/outlet pairings were also fixed in this paper, whereas in Fandel et al. (2022), results with both fixed and random inlet/outlet pairings were compared. As described in Fandel et al. (2022), holding the inlet/outlet pairings fixed reduces the variability in the resulting conduit networks. The results presented in this paper are therefore most likely less variable but more realistic representations of the system than the results presented in Fandel et al. (2022). The pyKasso-generated probability map of conduit locations in both cases closely matched the

branching pattern of the expected network (Figure 3a), despite the limitations of using a low-resolution, simplified, 2D geologic model as the travel medium and a proxy for the hydraulic gradient. These results support placing confidence in the ability of pyKasso-generated ensembles to provide information about the approximate configuration of karst conduit networks in this system. Displaying the ensemble of simulations on the same map also provides information about the probability of conduit occurrence in different zones of the study area: zones where most of the simulations generate conduits in the same general

location and orientation suggest that the probability of a real conduit existing in that location and orientation is higher (e.g. conduits departing from inlet N16 in Figure 3a), while zones where there is a wide spread of simulated conduit locations and orientations indicate that lower confidence should be placed in model predictions for that part of the study area (e.g. conduits departing from inlet N6 in Figure 3a) .


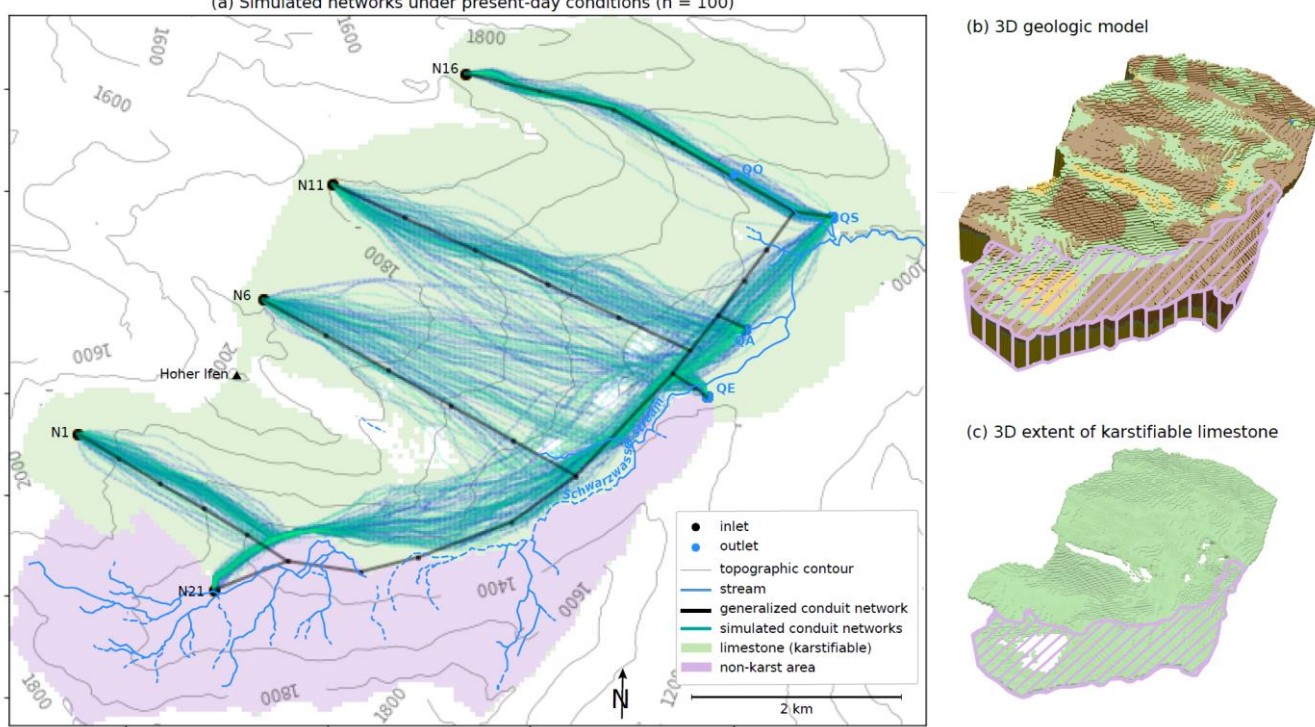

**Figure 3: Conduit simulations under present-day conditions. (a) One hundred simulations with inlet/outlet pairings fixed based on tracer test results (Goldscheider 2005). Stochasticity arises from the fracture network. The simulated conduits generally match the expected network configuration, but conduit location and orientation predictions are more certain in some areas than others (e.g.**
**high certainty departing from inlet N16, low certainty departing from inlet N6). (b) 3D geologic model with overlay showing the non-karstifiable zone of flysch and other units. (c) 3D extent of karstifiable limestone unit only, used to create 2D projection used in (a).**

## 5.2 Hypothesis testing

To test the two hypotheses for the past geologic conditions controlling conduit formation, 100 network realizations were run for each scenario using pyKasso, at the same model resolution previously used to simulate present-day conditions (50 m x 50
m cells).

To simulate Hypothesis 1 (glaciation closing all springs except QO), the model cells covered by the glacier were assigned a very high travel cost, discouraging conduits from crossing the glaciated area (Table 1). The inlets and outlets covered by the main glacier and the two smaller valley glaciers (inlets N21 and N16 and outlets QE, QA, and QS) were removed from the

model, and inlet N16 was replaced with an inlet at the entrance to the Hölloch cave. Under this scenario, inlet N1 was separated from the rest of the karst system by the glacier, and was assumed to have drained to a local spring with unknown exact location that is no longer active today (QI). However, the extent of the glacier in this subcatchment is uncertain, so the focus of this simulation remains on the eastern portion of the karst network connected to the paleo-spring (QO), which drained all remaining inlets not covered by the glacier (Figure 4a).

To simulate Hypothesis 2 (QS is much younger than the other springs), the Sägebach spring was removed from the list of system outlets and replaced with the paleo-spring (Figure 4b). The existing inlets remained the same as the present-day configuration. Under this scenario, the Aubach spring (QA), which is lower in elevation than the paleo-spring, was assumed to have served as an "attractor" receiving flow from the entire system, including the Mahdtal valley. An additional inlet was therefore co-located with the paleo-spring and assigned to the Aubach spring.

For both scenarios, the inlet/outlet pairings and iteration order were kept constant. A new fracture network was stochastically generated for each realization, using the same input statistics as the simulations of the present-day network. All other parameters were held constant.

**Table 1:** Travel costs assigned to different geologic settings. Higher travel costs discourage conduit formation.

| Geologic Feature | Travel Cost |
|---|---|
| Inactive model cells | 0.999 |
| Karstifiable limestone unit | 0.3 |
| Non-karstifiable rock units | 0.6 |
| Glaciated zone | 0.6 |
| Faults | 0.2 |
| Fractures | 0.2 |
| Conduits | 0.1 |

## 6 Findings

The modeled probability maps support Hypothesis 2 (a comparatively young QS) more than Hypothesis 1 (glaciation) (Figure 4). The simulated networks under Hypothesis 1 (glaciation) only matched the location and orientation of the mapped conduits in the active upper portion of the cave system, not in the inactive lower portion which is the focus of inquiry (Figure 4a). Of note, the spread of predicted conduit locations in this area includes a wide range of possible conduit locations northwest of the mapped cave system, but out of the 100 simulations, not a single one predicted conduits located along the axis of the actual cave system. These results support the conclusion that Hypothesis 1 has a low probability of explaining the formation of the inactive conduits.

The simulated networks matched the location and orientation of the mapped cave system better under Hypothesis 2, when the Sägebach spring was removed from the list of outlets and replaced with the paleo-spring. The orientation of the simulated

270   conduits is tightly clustered and matches well, while the location of the simulated conduits is slightly to the east of the mapped conduits with more overlap in the southern portion (Figure 4b).

Under this scenario, in the present day, the partially-mapped inactive conduits should extend to make a connection from QO to QA. Because QO is at a higher elevation than QA, the general flow direction when these conduits were active would have been from QO towards QA, even if the inclination of some individual conduits is inverse. Although these conduits now are

275   generally dry, it is possible that under extreme high-flow conditions, groundwater in the Mahdtal syncline overflows the southward bordering anticline and flows towards the Aubach spring (QA).

While these results indicate that Hypothesis 2 is a more likely representation of the past geologic conditions that led to the formation of the inactive portion of the Hölloch cave system that Hypothesis 1, other explanations cannot be ruled out. The primary contribution of the model results is that Hypothesis 1 can be ruled out as significantly less probable than originally

280   expected based solely on tracer tests and hydrogeological intuition.

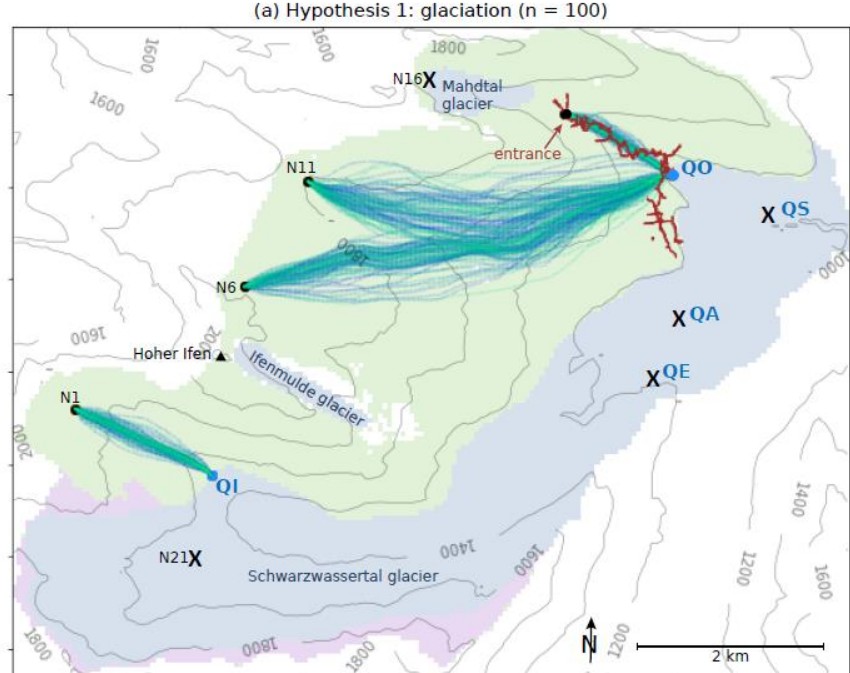

(a) Hypothesis 1: glaciation (n = 100)

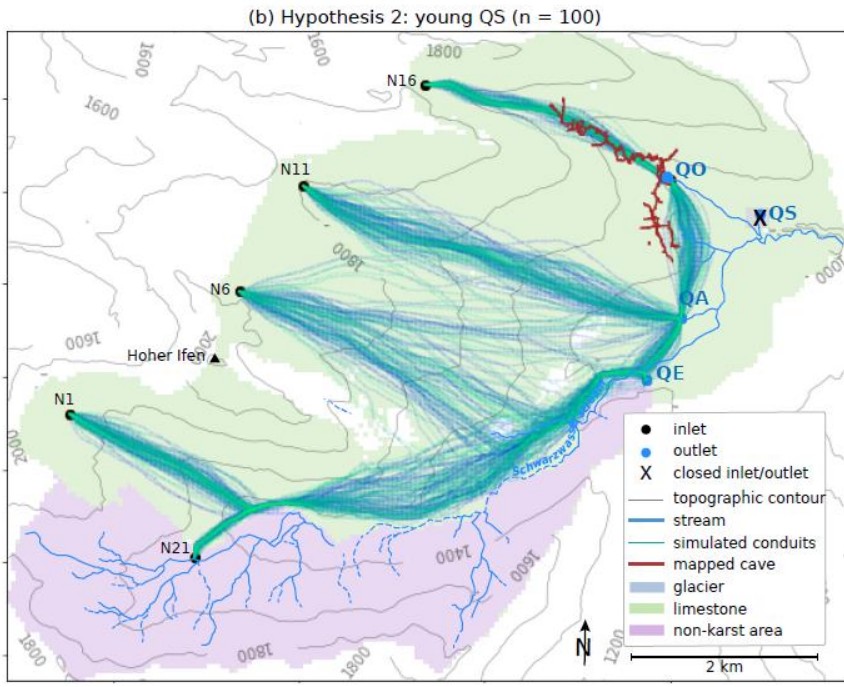

(b) Hypothesis 2: young QS (n = 100)

**Figure 4: Probability maps for 100 simulations under each hypothesis, compared to the observed cave map. (a) Hypothesis 1 yielded no networks matching the mapped cave system. (b) Hypothesis 2 yielded networks matching the orientation of the inactive cave network, located slightly to the east of the mapped network. This suggests that hypothesis 2 is less probable than hypothesis 1.**

**7 Discussion**

The model ensembles in this study clearly support one hypothesis over the other: the probability map for Hypothesis 1 shows a range of possibilities, but no conduits along the mapped portion of the inactive cave system, while the map for Hypothesis 2 shows a cluster of conduits suggesting that the inactive portion of the Hölloch cave continues southeast for several hundred meters before turning first southwest, then due south to connect to the Aubach spring (Figure 4b). This configuration is illustrated schematically in Figure 5.

These results suggest two additional ways that these hypotheses could be further tested by fieldwork:

1) Additional speleological explorations, focusing on the passages trending in the directions projected by the probability map – if the hypothesis is correct, these passages will continue rather than terminating in dead-ends. This information may be helpful in guiding future speleological exploration.

2) A tracer injection under extreme high flow conditions at the Hölloch cave entrance in the Mahdtal valley with sampling at the Sägebach spring (QS) and the Aubach spring (QA). When, under high flow conditions, the water level in the cave system is greater than the elevation of the Aubach spring (QA), the paleo-spring (QO) becomes active. Previous tracer tests under normal flow conditions found a connection only to the Sägebach spring, but under extreme high flow conditions, the normally inactive conduits projected to connect to the Aubach spring may reactivate. Unfortunately, the logistical challenges of waiting for such an extreme event to occur and then achieving a high-quality tracer test on short notice are significant.

A third hypothesis is also possible, a combination of the two scenarios explored in this study: that an initial phase of conduit formation occurred dominated by the influence of the glacier covering the valley, another phase occurred after the glacier had retreated but before the Sägebach spring was exposed, and the most recent phase developed with all three major springs (QE, QA, and QS) active. If this were to have occurred, two sets of inactive conduits should be present, corresponding to the two past phases of karstification. Currently, no conduits have been found along the paths expected if glaciation had influenced conduit formation. However, exploration of the Hölloch system is incomplete. If future exploration reveals new conduits along the paths simulated under the glaciation scenario, this hypothesis of multi-phase karstification could also be supported.

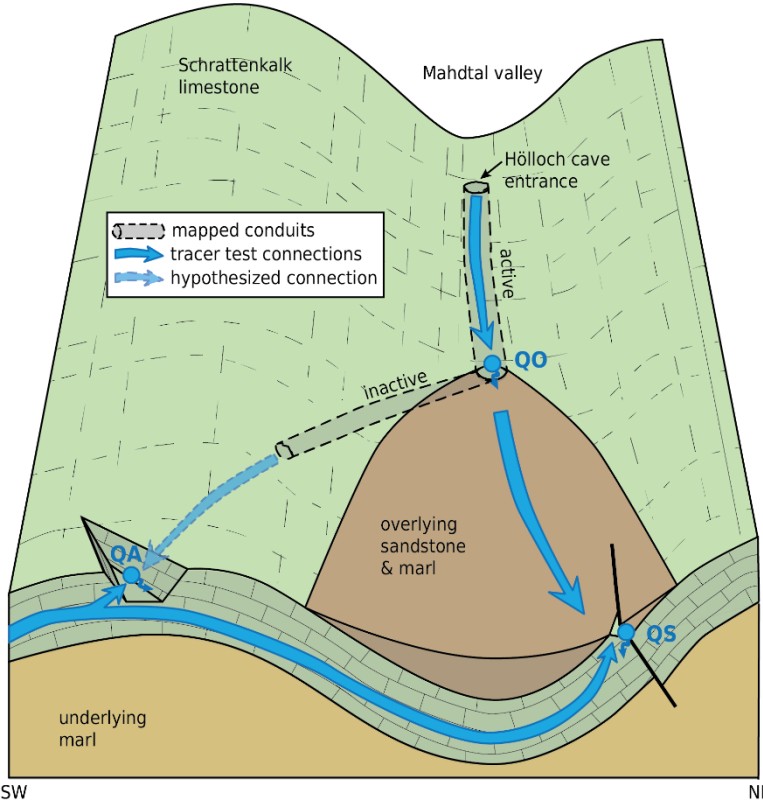

**Figure 5:** Schematic block diagram of the northeastern portion of the study site based on Hypothesis 2, showing the relationships between the subsurface conduit system and the points where it connects to the land surface. Surface water can enter the system through diffuse recharge where karstified limestone is exposed at the land surface, as well through concentrated point recharge through vertical shafts (e.g. the Hölloch cave entrance). Groundwater can exit the system at three springs: QA (large, well-developed), QS (limited discharge capacity), or QO (only during extreme high-flow conditions). The karst system is confined by overlying impermeable sediments in the lower part of the Mahdtal valley. An active cave stream flows through subsurface conduits from the Hölloch entrance towards QO. Flow connections from the entrance to QS, from QO to QS, and from QA to QS, have been demonstrated by tracer tests. Inactive conduits to the southwest of QO have been partially explored. Modeling results suggest that these conduits continue, connecting QO and QA.

While the model results provide insight into this area's geologic past, some questions and limitations remain. In both scenarios, the underlying geologic model is a limitation. A 2D geologic map simplifies the complex three-dimensional reality. However, a preliminary test using a development 3D version of pyKasso, with the 3D geologic model as input, yielded conduit maps that did not appear significantly different from the 2D version used in this paper.

The simulations representing each scenario were generated based on the same set of model parameters (chosen based on their ability to reproduce the present-day network), with stochasticity arising from the fractures and hierarchical approach. This approach was chosen to minimize the number of variables under consideration. However, it may limit the range of network configurations generated. For example, with the chosen parameters, more direct connections between inlets and outlets are somewhat favored, whereas with different parameters, paths following the anisotropy gradient or paths following fractures might have been more favored, even if they are longer. Different parameter choices could potentially have led to different

results for the scenarios considered – for example, under Hypothesis 1 (glaciation), parameters de-emphasizing the importance of choosing a more direct path could have led to conduits that descended the gradient more directly until they encountered the edge of the glacier, then turned northeast to develop along contour, more closely matching the mapped inactive conduits. This highlights the importance of designing modeling experiments carefully and of choosing a consistent parameterization strategy *before* actually running the simulations.

One limitation of this study is that the order of events and the rate of change in the landscape is not precisely known. Field observations suggest a late glacial to postglacial age of the Sägebach spring and a relatively rapid exposure. The lack of precise dating limits our understanding of the system as a whole. Another question arises from the observation that, although the simulated conduits under Hypothesis 2 (a young QS) are much more similar to the inactive portion of the Hölloch cave than under Hypothesis 1 (glaciation), they do not exactly match the mapped cave system. The simulated conduits tend to lie slightly to the east of the mapped conduits, skirting the edge of the overlying sandstone unit. This is likely due to our simplification of the geology – in reality, the karstifiable limestone in this part of the system is covered by non-karstifiable sandstone, marl, and quaternary sediments. These overlying units may have made it more difficult for conduits to develop beneath them, encouraging conduit development along their northwestern boundary. However, the mismatch may also indicate a conceptual gap in our understanding of the system.

A few examples of factors not considered in this study that could affect conduit formation include:

1)      The presence of an unmapped tectonic feature with higher or lower conduit-forming propensity than the surrounding rock. This would attract or repel conduits in the model.

3)      An eastward (downgradient) shift in the location of the contact between the karstifiable limestone and the thin overlying sandstone unit as the sandstone erodes over time. The modeled conduits in this specific location tend to follow the contact, so the real conduits being slightly westward of the modeled conduits could be explained if the contact were previously also slightly westward of its current location.

4)      More phases of karstification than hypothesized could have occurred, in various orders, resulting in a conduit network that is partially but not fully explained by the hypotheses presented here.

**8 Conclusion**

This study demonstrates the application of anisotropic fast marching methods for rapid model ensemble generation to test competing hypotheses describing the past geologic conditions that controlled karst conduit evolution in a real catchment. It extends the applicability of these methods from situations where the goal is to generate a map of an unknown conduit network, to situations where the goal is to understand how an already-mapped conduit network was formed.

For the field site investigated in this study, the Gottesacker karst system, comparing the conduit network under different past scenarios (based on model-generated probability maps of conduit locations) to the actual conduit network (based on maps of the explored present-day cave system) allowed the identification of the past conditions with the greatest influence on cave

development. The model results indicate that the most probable scenario is that the conduits of interest formed before the lowermost spring in the catchment came into existence, and drained to the other two major springs in the system (as opposed to conduit formation being dominantly influenced by glaciation occluding all three major present-day springs). The modeling results also enabled making recommendations to guide cave exploration efforts and future data collection that could further test the conduit formation hypotheses.

The strength of the anisotropic fast marching method is its comparatively low cost in terms of computational resources and quantity of initial input data required. This allows for rapid iteration and exploration of many different scenarios, which makes it especially well-suited for ensemble modeling. The model ensemble approach demonstrated in this study could be applied to other karst systems with mapped conduit networks, both to better understand the past geologic conditions that influenced the conduit network development, and to better target future mapping and data collection efforts to answer outstanding questions about the system. It also holds potential for exploring competing conceptual understandings of a karst system, and identifying possible conceptual gaps, both of which are significant sources of uncertainty in predicting karst systems' response to anthropogenic stresses.

**Code availability**

The code and supporting data used to generate the figures in this paper are presented in a Jupyter Notebook available in a public GitHub repository here:

https://github.com/randlab/pyKasso/blob/c295727053c51d9f4ba735a171a5e94df4e1f48a/notebooks/Fandel_et_al_2023_HESS.ipynb

**Author contribution**

NG developed the concept for the paper and the hypotheses, and provided data on the study site. FM, CF, and PR developed and debugged the pyKasso modeling package. CF implemented the model and performed the simulations. TF provided research supervision and guidance in the use of model ensemble approaches. CF created the figures and wrote the manuscript draft, and NG and PR reviewed and edited the manuscript.

**Competing interests**

The authors declare that they have no conflict of interest.

## Acknowledgements

The authors wish to thank the many speleologists of the Höhlenverein Sonthofen who contributed to mapping and digitizing the Hölloch cave system, particularly Andreas Wolf from the German Institute for Karst and Cave Science. This research would not have been possible without their exploration efforts and generosity in sharing data.

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
