# Peer review of "Improving understanding of groundwater flow in an alpine karst system by reconstructing its geologic history using conduit network model ensembles"

_Hydrology and Earth System Sciences, 2022_

## Author Response (AR1)

**Response letter for HESS-2022-280**

**Title:** Improving understanding of groundwater flow in an alpine karst system by reconstructing its geologic history using conduit network model ensembles

**Authors:** Chloé Fandel, Ty Ferré, François Miville, Philippe Renard, Nico Goldscheider*

**Editor's comment:** On the basis of the Reviewers' comments and the Authors' responses and arguments, I would be willing to give the Authors a final opportunity, with the understanding that an additional round of reviews will be required. In case the revised study does not fully satisfy the Reviewers who will be involved, it will not be accepted.

**Response:** Thank you for accepting our revised manuscript. We have significantly changed the model inputs, reworked the text to provide more explanation where needed and to shorten where possible, as well as providing a new set of figures. Please see our detailed responses to reviewer comments below.

**Materials included in this submission**

1. The revised manuscript with markup showing the changes made
2. A point-by-point-response list (this document)
3. A clean version of the revised manuscript
4. New figure files

**Point-by-point response to referee comment 1:**

*Comment 1: In the document the authors state that the simulation of the networks for a resolution of 50x50 meters can be performed in less than 2 minutes. In this sense, what is the limiting factor for not using a map with a higher resolution?*

Response: Explanatory text has been inserted in section 5.1.

*Comment 2: In Figure 3, how is the expected network defined?*

Response: A more specific description of how the expected network was defined has been inserted in Section 3.

*Comment 3: In Figure 3, is it possible to draw the expected network with different plotting settings to enhance its visibility? As it is, the expected network can be confused with the simulated conduits.*

Response: The color scheme and plotting settings for all figures have been revised to make the geology, expected network, and simulated networks more visible.

*Comment 4: Could you provide an interpretation of the variability of the simulated conduits? Is it an indicator of the uncertainty of the expected conduit? Could you include a metric to measure such variability?*

Responses: We have added explanatory text toward the end of section 5.1. The main point is that variability in the simulated conduits is a result of the model's stochastic nature. Stochasticity can be introduced in several ways, but for this study, in the simulations of the competing scenarios under consideration, the primary source of stochasticity is the fracture network. For each conduit simulation, a new fracture network is generated, based on the descriptive statistics obtained from field and aerial photo mapping of the actual fracture network. The conduits then form preferentially along fractures, resulting in slightly different networks for each simulation. In regions of the model strongly controlled by fractures, there will be more variability in the ensemble in response to the different fracture networks in each simulation. In regions of the

model where other factors (such as gradient, existing conduits, or obstacles) are more dominant, there will be less variability in the ensemble. The "fuzziness" of the ensemble maps (more fuzzy = more variability in where different model simulations predict conduits will go) is an indicator of model uncertainty. Regions where the different model simulations all predict different conduit paths would be regions where we have low confidence in the model's predictive abilities, and where we would be interested in acquiring more data. Regions where the different model iterations cluster tightly around a single path would be regions where we have higher confidence that a conduit is indeed in the model-predicted location, and we would not prioritize collecting additional data. This is discussed in more depth in Fandel et al. (2021).

The probability maps of the frequency of conduit occurrence at different locations for all the simulations in an ensemble are a quantitative and visual metric of variability – pixels with darker coloring indicate a higher probability of a conduit being present in that pixel. The original maps showed the simulated probability in 1% increments (black cells had a 100% probability of having a conduit, transparent cells had a 0% probability of having a conduit). To improve visibility, we have changed the color scheme so that each network plots in a slightly different color (in the blue-green range) to facilitate identifying conduits that belong to the same simulation. We also binned the probabilities into 10% increments. In the current figures, the highest-color-saturation cells indicate a probability between 90 and 100 %, while the least-saturated cells indicate a probability between 0 and 10%. This increases the legibility of the maps. The black and white maps are still available if needed for further analysis.

We have been exploring other metrics to quantify the similarity or dissimilarity of conduit networks to each other, and do not yet have a simple satisfying answer. Quantifying the similarity or dissimilarity of conduit networks and their variability is challenging. One approach is to represent the networks as mathematical objects: graphs with nodes connected by edges. Statistics can then be computed describing the geometry and topology of these graphs. Collon et al. (2017) describe the statistics most relevant to describing karst networks, and we have incorporated their functions to calculate these statistics into the pyKasso model code. However, most of these statistics quantify differences in the topology of the networks and are more relevant to classifying different types of cave networks. When attempting to quantify the degree of variability in our simulated conduits, the models in our ensembles have similar structures (number of extremities, branches, junction nodes, etc.), so these statistics are not useful in this situation to represent the geometric variability in the ensemble. Identifying other statistical metrics that can usefully describe differences between models of the same network in an ensemble is an area of future research that we are highly interested in pursuing further, but we consider that our results are not mature enough and that further exploration of this topic is beyond the scope of this paper.

*Comment 5: In line 243 the authors mention that there is an additional scenario that is not explored in this study. Is there any justification for not analyzing it?*

Response: The scenario in question is that there were several overlapping phases of karstification, in which different combinations of springs were either exposed or occluded as glaciers went through cycles of growth and retreat. While the two scenarios we explored were supported by clear geologic, hydrogeologic, and geomorphologic field observations that allowed us to define clear, well-bounded hypotheses, this is not the case with the additional scenario – it is not well-defined enough to be testable. Furthermore, one of the two tested scenarios delivered cave patterns that match the observed cave pattern very well, suggesting that this is how the caves have essentially formed. We agree that the statement about the "additional scenario" in line 243 causes confusion and distracts from the logical sequence of our research work. Therefore, we will delete this statement from the revised paper, since it does meaningfully contribute to the hypothesis testing we are focusing on.

*Comment 6: In line 125 the authors wrote that in this study you are considering that conduits form preferentially in the direction of the maximum downward hydraulic gradient. Could you please add some references to studies where this assumption is also employed? Alternatively, could you please add some words explaining the reasoning behind this assumption?*

Response: This is indeed an important point. We have added explanatory text and additional references to explain the influence of both hyraulic gradient and bedding plane orientation on conduit formation. We have also changed the orientation data used in our simulations – in the revised simulations, we use the surface of the lower boundary of the karstifiable unit to determine the preferential orientation of conduit formation. Previous work by Filipponi (2009) indicates that conduits in this system are strongly controlled by bedding plane orientation. We have inserted explanatory text in Section 5 and 5.1.

*Comment 7: In Section 6 (Findings) you do some references to Figure 4. However, they seem to refer to elements in Figure 5.*

Response: Thank you for catching this mistake! We have consolidated Figure 4 and Figure 5 into a single figure in an effort to be more concise, and have updated the figure references accordingly.

*Comment 8: There is a missing closing bracket in line 112.*

Response: Thank you for catching this. We have re-written this section to address other comments, and have in the process corrected the missing bracket.

**Responses to referee comment #2:**

General note: Because this comment is quite long, we have condensed some the main points, then addressed each one.

*Comment 1: The lightness of the contribution as compared to the recent papers already published by the same authors on the same topic and the same case study. The submitted paper is a follow-up of two papers recently published in Hydrogeology Journal: Fandel et al. 2020 A model ensemble generator to explore structural uncertainty in karst systems with unmapped conduits and Fandel et al. 2022. The stochastic simulation of karst conduit network structure using anisotropic fast marching, and its application to a geologically complex alpine karst system. In this new submission, the concepts, tools, and data are identical to those presented in these two previous papers. The 2021 paper presented the application of the previously-published SKS approach to the Gottesacker case study. The 2022 used an anisotropic fast marching algorithm instead of the initial version that used an isotropic fast marching . In the present work, the two hypotheses for the formation of the inactive karst conduits are translated in the modelling approach as changes in the position and pairing of inlets/outlets. They are described in 22 lines page 9 + the figure 4. All the remaining of the methodology (page 1-8) is a rewording of the previous papers. As well as a large part of the discussion and of the "messages" of the paper. As the inlets-outlets pairing effect was already discussed in the previous 2022 paper, the example of a pairing guided by a speleogenetical history would have been more pertinent as 15 additional lines in the previous paper than as a "new" 16 pages paper, avoiding the large number of redundancies.*

Response: We have reduced the amount of text and have removed some of the figures which served primarily to explain our previous work. Because this paper is the third in a series, each of which builds on the preceding work, some minor repetition is necessary so that readers unfamiliar with the first two papers can understand the third. However, the work presented in this paper is completely novel and fundamentally different from the work presented in the previous two papers. The first paper was focused on developing modeling capabilities in the Gottesacker karst system, while the second one was focused on developing anisotropic fast marching methods for conduit simulations. This paper focuses on applying these methods to understand the geologic history and conduit evolution of the Gottesacker karst system. This is the first time that a stochastic conduit simulation algorithm has been applied to questions about past geologic conditions. Usually, simulations focus on projecting conduit maps based on the geologic setting, whereas here we attempt to reconstruct the geologic evolution based on conduit maps. This is also only the second application of anisotropic fast marching algorithms for conduit simulation. To our knowledge, no other published work

uses anisotropic fast marching algorithms to simulate conduits, nor does any published work use conduit simulations to understand past geologic conditions.

*Comment 2: Over-exploitation (or even mis-exploitation (?)) of the modelling approach. The inactive conduit is oriented north-south. The modelling algorithm searches a path between an inlet and an outlet, with a secondary influence of fractures (randomly generated) and a pseudo-hydraulic gradient. If you do not put an inlet / outlet on the south of the paleo-spring, there is no reason (theoretically and numerically) to generate a path aligned with the inactive conduit: from top view, the path "south -> QO spring" is opposite to the input (= topographic) gradient. No need for the model to see that. Thus, as QO is the only outlet in hypothesis 1, only an inlet located approximately in its south could explain such a path. As the test performed by the authors does not propose this solution, it is obvious that they are not going to generate a consistent solution. In hypothesis 2, some paths are generated accurately, but it is because the authors let the spring QA exist: thus, some paths are generated between QO and QA which is a main direction aligned with the inactive part. Again, this is consistent with the topographic gradient and the direction QO-QA, no need for the model to guess this hypothesis is consistent with the observations. But, and this is really important, what about the direction of fluxes, never discussed (a limitation due to the fact they simplify in 2d)? Indeed, in hypothesis 2 these paths are obviously going from QO (higher) to QA (lower): in that case, QO is not the spring of the inactive part but an "entrance". What about the field data? Are the inactive conduits indeed sloping towards QA or in the opposite direction? Were there indices of QO being an inlet and not an outlet? It is not said so, in the text, where QO is presented as an outlet. If the conduits are sloping towards QA, then QO cannot be their previous outlet, and the tests performed for hypothesis 1 were, from start, bound to fail (again, no need to perform any computation to conclude that). If, oppositely, the inactive conduits are sloping towards QO, then their "dead-end" should be the place of a past inlet. And paths will be generated between them and QO, independently of the presence or not of the paleo-glacier (thus independently of both hypotheses). Most of all, it will imply that what the authors consider as a proof for benefit of hypothesis 2 is wrong as hypothesis 2 generates conduits sloping in the opposite direction.*

Response: Thank you for pointing out that we have not provided enough explanation of these questions. We have inserted explanatory text in the site description and in the section where we introduce the two main hypotheses to better describe the hydrogeologic setting of the field site and the field observations that led to the development of the two hypotheses under consideration. We have added a new block diagram to better explain the geologic setting. We have also revised the input data to the simulations, using the surface of the lower contact of the karstifiable unit instead of the land surface topography to guide the orientation of conduit formation.

The main points, which we elaborate in the text, are as follow:

1) There is not enough information to predict which path conduits will take without actually running simulations. The results could have been that none of the scenarios we ran produced conduits similar to the mapped inactive passages, or that both scenarios did, or that one did and one did not (this third possibility is what indeed occurred). The overall orientation of the mapped portion of the inactive conduit network roughly parallels a contour line. This is part of what makes it difficult to predict what conditions led to its development. In detail, the actual conduits exhibit a relatively complex three dimensional pattern, with ups and downs, siphons and shafts, and ramiform branchings. QO is higher than QA, so the dominant overall trend is that the conduits are sloping slightly from QO down to QA.

2) Because the slope of the conduits from QO to QA is gentle (approximately 100 m elevation change over 2 km distance) it is possible for these conduits to have formed in response to unusual hydraulic gradients, generated by groundwater "backing up" when it reaches the edge or base of the karstified zone. In the scenario envisioned in Hypothesis 1, the development of karst is limited beneath the glacier, such that water draining along the axis of the syncline immediately to the southwest of the Mahdtal valley could be funneled along this synclinal axis until it hits the edge of the glacier, and then back up and flow northeast, roughly along contour, until it finds an outlet (QO). It was not possible

to know in advance of running the simulations whether this would be a more or less cost-efficient path than if the conduits cut across diagonally in a more direct west to east line. If it had been possible, as described in this comment, to determine the influence of glaciation on conduit formation only from inlet/outlet locations and topography, there would have been no motivation to write this paper.

3) The flow direction of water is not solely controlled by the slope of the conduits – it is controlled by the direction of the hydraulic gradient, particularly in confined settings like this one. QO is an overflow spring – it has only been observed to flow once in recorded history, during extreme high-flow conditions, when all conduits beneath it were filled and "backed up" and could not accommodate any more flow, resulting in a reversal of the usual hydraulic gradient and overflow exiting the conduit system at QO. This can occur even if the conduits are sloping away from QO.

4) There is clear geomorphologic evidence that QO was an active inlet in the past. In the present day, there is very little surface runoff generated in its surface catchment area, so it no longer acts as an inlet except during periods of rapid snowmelt or intense rainfall.

*Comment 3: By definition, stochastic simulations are stochastic, not determinist. Thus, they do not aim to find the "true location" of a conduit but to propose equiprobable paths given a limited knowledge. In the present approach, the paths are found by fast marching. The cost is a combination between aerial distance (in 2D here) and topographical gradient (because it is what is used here), and fractures. Only the fractures change from one simulation to another, thus, it just allows to create a kind of a glow around the two main elements of the cost function: gradient and distance between the two points of the considered pair. Thus, as soon as a hypothesis is consistent with these two elements, no further ensemble simulation is required. The stochastic approach does not seem to provide any particular interest in that precise case.*

Response: This comment parallells reviewer 1's comment #4 inquiring about the source of the variability between simulations. We have added explanatory text about fractures, stochasticity, and confidence in response to both comments. They key point is that the value of the stochastic approach is in providing a sense for how consistent the ensemble of stochastic simulations are in their predictions of conduit locations. If most of the simulations agree in predicting a conduit in a certain location, we can be more confident in the likelihood of a conduit actually being present in that location, whereas if the simulations produce a wide range of possible paths between two points, we are less confident in identifying where an actual conduit might be present. In simpler terms, the degree of "glow" carries useful information. For example, we are much more confident in the path of the connection between QO and QS than in the path between QA and QS.

*Comment 4: The 2D approach simplification seems not appropriate as it completely ignores the staggering of the karstic conduits (illustrated in theory by the authors in their figure 1). The Borghi et al. algorithm was however 3D and other works by other authors proposed 3D approaches, why not having worked with them? In this precise case, the topographic gradient is used to mimic the hydraulic gradient, but what does the 2D maps represent? Do the authors consider that the conduits are vertical until they reach the 2D map level? In that case, should the paths computed by the model be seen as conduits developed along the piezometric surface? If yes, which one: today's surface or a paleo-water table? The paper is clear: the authors use only something which they associate to today's surface, but it is not consistent with their own initial illustration in figure1. If there is a scenario of water-level dropdown like the theory in figure1, this level is moving through time, and thus the "2D equivalent level" (and map) is changing… This is not at all considered by the model here. Another critical point related to this : if what we see in 2D maps are the conduits developed inside the limestone level after the conduits have vertically crossed the highest part to reach the water-table, then on this virtual surface, everything should be limestones => the geological units presented on this 2D maps have no meaning, more precisely, the sandstone and overlying units should be ignored: they have been crossed by the conduits vertically and on the virtual surface we consider, which is below, we are always inside the limestones (except when the limestones are completely eroded). Saying the same differently: the 2D units presented on the maps seem to be what is seen at the surface, while the network develops at depth, inside the Schrattenkalk. This problem was already there in the previous papers.*

Response: We have made several significant changes to address these comments – partly inserting additional explanatory text, partly changing the model input data.

pyKasso is based on SKS and was developed by the same research group. SKS, while 3D, had some serious limitations because it was only able to calculate isotropic fast marching, which yielded unrealistic conduits for this study site. pyKasso is the most advanced option currently available. A stable version of pyKasso is currently available in 2D. The 3D version is functioning but has not yet been thoroughly tested. We have run preliminary 3D simulations, which yielded conduit networks similar to those in the 2D simulations. This is not surprising, because the karstifiable limestone unit is relatively thin (~100 m), so the vertical range in which conduit formation is possible is small compared to the lateral extent of the karst system. These 3D simulations do not add significant additional information concerning the hypotheses tested in this paper. We have added a sentence explaining these additional simulations in the discussion, and are confident that the 2D simulations are relevant and sufficient for this particular study question.

We partly agree with the reviewer that using topography and a surficial geologic map was not the best way to construct the model. To improve the simulations for this study, we have changed the geologic map and the surface used to calculate the orientation of preferential conduit development. This is now explained in section 5.1 and Figure 3, as well as in our response to Reviewer 1's comment 6. Essentially, the new geologic map is a 2D projection of the 3D geologic model, showing all locations where the karstifiable limestone unit exists (i.e. has not been removed by erosion), and where the non-karstifiable flysch zone is located (see Goldscheider 2005 for a detailed explanation of the flysch zone – in this part of the catchment, only surface drainage occurs and no karst is present). We also now use the lower boundary of the karstifiable unit as the surface used to guide conduit orientation, rather than using the land surface as before.

*Comment 5: Figure 1 is misleading: it is presenting the principles of a two-phase karstification due to a drop in the base level, resulting in a two-level network. But the used approach in the paper is 2D and can not consider different levels of karstification as elevation is not taken into account in a map view. It is indirectly considered with the anisotropic fast marching approach, where only a global gradient, here parallel to the topography, is introduced to try mimicking the gravity effect. Using such a figure is not appropriate if it does not correspond to the spirit of what is feasible with the approach and if it does not correspond to what is tested/investigated in the following.*

Response: This figure was indeed not a good representation of the processes at work in this example, and was merely showing a generalized example of how two-phase karstification could occur. As it is not necessary to the main points of the paper, we have removed it and instead included a block diagram of our area of focus, the Mahdtal valley.

*Comment 5: Section 2, section 4 and section 5.2 almost say differently the same things: the hypotheses should not be repeated and split in various incomplete parts but regrouped in a single complete section.*

Response: We have added a significant amount of text to section 4 describing the geological evidence for the two hypotheses. Section 5.2 describes instead the model implementation of these hypotheses. Section 2 is a brief summary of the paper overall as an introduction for readers.

*Comment 6: Line 53: "this study presents a model-based approach to reconstructing the geologic processes driving cave formation". I disagree, the modelling approach builds networks by considering the influence of user-defined geologic influencing factors. It does not reconstruct a geologic process, as no physical rules/equation are used.*

Response: We completely agree that the approach presented here is not a process-based speleogenetic model, and was never intended to be. This is discussed in our previous paper, and in section 5 of this paper, so to minimize repetition we do not discuss the topic further in this paper. We have rephrased this sentence to

make it more clear that we are attempting to identify which processes were most determinant in the cave forming where it did, rather than trying to simulate those actual processes.

*Comment 7: Line 57: why only two hypotheses? Why only the QS being younger? What about the "ages" of inlets? Inlets could also change depending on the erosion of overlying deposits…*

Response: See response to comment #3, and to reviewer 1's comment #5. We have added a significant amount of text and a block diagram explaining the geologic evidence for these two hypotheses. While many other explanations could exist, we do not have sufficient information to frame them into testable hypotheses. The main contribution of this study is really in that it provides an argument for eliminating one of the two most well-supported hypotheses (glaciation). We have modifed the conclusion to reflect this.

*Comment 8: Figure 2: use different colours for active vs. inactive conduits.*

Response: This is quite difficult to accomplish (it requires three separate steps in three separate software programs), so unfortunately we were not able to make this change within the time available for revisions. We agree that it would be optimal, but we do not find it essential.

*Comment 9: Currently and considering the conceptual schema of Chen et al., QO is linked to QS: on the figure it is only a stream: what happens at depth? Do we have information about the connections between the QO point on map and QS point?*

Response: See additional explanations of field observations from response to comment #3.

*Comment 10: The consistency between the 2D map and schematic cross-section is not clear : contrary to the map, on the cross section, the marls do not outcrop in the valley, as well as the sandstone on the Hoher Ifen, the marls at the same point, then again the sandstone at Gottesacker, etc… Looking at the geological map provided in the 2021 paper, the 2D map used here is also not consistent : limestones should outcrop almost everywhere. The geology is not complex, and the rasterizing effect does not justify all these differences. As this seems to be used in the cost function, this is questioning. On the cross section, N1, N6, N11 and N16 should be indicated to help the reading.*

Response: See responses to comment #4. We have changed the geologic data used as a model input.

*Comment 12: Holloch cave: show developed cross-sections of the conduits if possible (vertical organisation?*

Response: This again is a time-intensive request that we agree would be beneficial, but not essential. We did not have time within the response window to include these plots. Detailed vertical cross-sections are available in the publication describing the explored cave network: Höhlenverein Sonthofen: Das Hölloch im Mahdtal – 100 Jahre Höhlenforschung im Kleinwalsertal, Sonthofen. Sonthofen, 2006.

*Comment 13: Line 107: hypothesis 2 not clear: if the connection is from QO to QA, what should today explain the dis-activation as QA still exist?*

Response: See explanatory text in response to question #3. The paleo-spring is only active today when the hydraulic gradient reverses in response to extreme high-flow conditions.

*Comment 14: Line 127: in 2022 paper, the hydraulic gradient was said to coincide with the elevation of the bottom surface of the limestone for the synthetic case. Here, it is said it is the topography, with the justification "it is simpler to calculate" (line 123): how do you explain such differences? The cross-section in figure 2 is not so evident with different level of erosion for limestone (the Hoher Ifen for example and its surroundings). "Simpler to calculate" is not relevant if the consequences are large (see remarks in above comments).*

Response: This is indeed an important point. See response to comment #4 and to reviewer 1 comment #6. We have changed the model input to calculate the gradient from the bottom surface of the limestone and added references supporting this assumption.

*Comment 15: Line 136: "The conduits leading to each outlet can be simulated in separate iterations, to represent springs of different ages": in SKS there are not 1 iteration but several ones as you simulate one path between 2 points, and then a second one which is influenced by the first one (second iteration). The authors explain it well in their first paper, thus here this sentence is unclear: what was the real message to pass?*

Response: Thank you for pointing this out. We have removed this sentence and replaced it with a reference to the previous paper.

*Comment 16: Section 5 and 5.1: a resume of previous papers (but see point 1 of the main comments).*

Response: We have made significant changes to the simulations in response to reviewer comments, and now describe these how these changes differ from the previous simulations in section 5 and 5.1.

*Comment 17: Line 155: referring to Fandel et al 2021 for a detailed description of geology is strange (and partly unfair) as the geology was there taken from the previous works of Goldscheider et al., Chen et al., etc.*

Response: The geologic and karst hydrogeology setting was indeed first thoroughly described by Goldscheider et al. The first groundwater flow model was described by Chen et al. These excellent papers are the foundation for our current work, and are cited with appreciation in Section 3 describing the field site. However, these papers described the *conceptual* understanding of the geology. Line 155 is referring to the first 3D geologic *computer model* of the site, created in GemPy by C. Fandel based on the conceptual understanding and data from these earlier papers. This computer model was first presented in Fandel et al. 2021, which is why that is the paper cited in this line. Therefore, no changes have been made to this line.

*Comment 18: Line 157: the model is said "sliced using the topography surface": why that? the karstic system does not develop at the topographic surface (see also detailed comment above).*

Response: See response to comment #14.

*Comment 19: Line 158: cells of 50x50m seem quite big for the level of detailed searched here… (the difference in source elevation is largely lower which can induce large border effect in the method, see previous comment on the 2D approximation).*

Response: See response to reviewer #1 comment on resolution.

*Comment 20: Line 161-171: all already said in previous papers.*

Response: See previous comments. Some repetition of previous papers is needed for readers to be able to understand the current paper.

*Comment 21: Line 163: "expected paths". These paths are not the karstic conduits in itself (the superposition of the Holloch cave map show this clearly) but a equivalent conceptual model allowing to reproduce the hydraulic response. Why do the authors try to fit it instead of the conduit real paths if they have them? Also, if we have not the real paths, as the concepts guiding the conduit modelling approach is not the same than trying to find an equivalent hydraulic model, why trying to compare both model results? They do not have the same purpose.*

Response: See response to Reviewer 1's comment about how the expected network was generated. Explanatory text has been added. The expected paths, while not mapped directly through speleological exploration, are the best available approximation, based on a significant amount of data. We feel confident that this expected network represents the general locations and orientations of the actual active conduits. Since our study questions focus on general locations and orientations rather than detailed maps, the expected network map provides an appropriate level of information for our study.

Detailed maps from speleological investigations are only available for the small portion of the network in the lower Mahdtal valley. Since our analysis includes conduits outside these segments, the speleological maps are insufficient for our purposes.

*Comment 22: Line 169: "These results support placing confidence in the ability of pyKasso-generated ensembles to simulate": I find it an "over-interpretation". The end of the sentence: "particularly when the inlet/outlet assignments are fixed" confirms what I said above about the importance of inlet/outlets pairing issue.*

Response: We have modified this paragraph to attempt to clarify what information our simulations can and cannot provide.

*Comment 23: Figure 3: Already there in Fandel et al 2022 (said by the author in the legend): why spending so much time on already presented works and results?*

Response: In an effort to reduce repetition, we have replaced Figure 3 with a figure showing the conduit network ensemble generated for this paper, using modified inputs to address other reviewer comments, and have shortened and modified the accompanying text to focus on the simulations presented in this paper, rather than our previous paper, and directed readers to our previous paper for more details.

*Comment 24:*
*Line 192: what proves that the other inlets N1, N6 and N11 exist at the time of the glacier? Is there any field element for that?*
*Line 194: "The existing inlets remained the same": why, what support this assumption?*

Response: See response to earlier comments about field evidence for hypotheses. We have added a significant amount of text describing the field evidence supporting our understanding of which inlets and outlets were active at different times.

*Line 251-257: globally break open doors.*

Response: This sentence is not found anywhere in our paper. Perhaps this is a typo?

*Line 261: "This is likely a limitation of the model's ability and our simplified assumptions to predict exact conduit locations rather than general orientations and connections" As I said in the above comments, the goal of stochastic approach is precisely NOT to predict the exact conduit location. If you want an exact prediction, you have to use a deterministic approach. I am surprised by this sentence.*

Response: We have significantly edited this section to address the changes resulting from our updated model inputs.

*Section 7.1: Not very informative, it is globally a rewording of what was previously said.*

Response: Recommendations for how to shorten papers are always most welcome! In an effort to be concise, we have removed this section.

---

## Author Response (AR2)

**Response letter for HESS-2022-280 Round 2**

**Title:** Improving understanding of groundwater flow in an alpine karst system by reconstructing its geologic history using conduit network model ensembles

**Authors:** Chloé Fandel, Ty Ferré, François Miville, Philippe Renard, Nico Goldscheider*

* corresponding author

**Editor's comment:** The results of this second round of reviews enable me to suggest an additional round of very careful revisions. These should include an appraisal of the originality of the anisotropic fast marching, considering the context tackled, as well as a detailed assessment / re-evaluation of the interpretations offered for the modeling results. It is also not my intention to discount any of the constructive comments emerged during the revisions. The revised manuscript will then undergo a very last round of reviews.

**Response:** Thank you for accepting our revised manuscript. We have included information for the editor regarding the question of the originality of anisotropic fast marching, as well as a reassessment of the interpretation of the modeling results. Please see our detailed responses to reviewer comments below.

**Materials included in this submission**

1. The revised manuscript with markup showing the changes made
2. A point-by-point-response list for the reviewers
3. A clean version of the revised manuscript

**Point-by-point response to referee comment 1:**

*Comment: I am satisfied by the answers provided by the authors. All the comments/suggestions are correctly addressed in the revised version of the manuscript.*

Thank you for this clear and concise assessment of our work.

**Point-by-point response to referee comment 2:**

*Comment 1: The authors have made substantial changes in this new version that are going in the right direction. The new paper is more concise, more focused on the contribution of this work, which is better explained. However, some modifications are still required before considering it for publication.*

Response: Thank you for taking the time to review our revised paper. The requested modifications are discussed point-by-point below.

*Comment 2: Concerning the contribution, the authors now better frame it. However, they still insist on the fact that they are the first to use anisotropic fast marching, which is not true: Luo et al. published in 2021 (in Journal of Hydrology) a paper proposing to use anisotropic fast marching (AFMA) also in the frame of the algorithm by Borghi et al. (2012) and also for stochastic karst network modeling. Luo et al. use the algorithm in 3D, and exploit the AFMA to render the effect of fracturation without explicitly simulating the fractures, which is very interesting (I specify that I am not in this team, nor related to it). This is a different exploitation of AFMA that the one proposed by Fandel et al., which uses AFMA as a way to render the effect of "bathymetry of the inception level" in a 2D model. As well as the previous*

*papers should have better presented the alternatives to Borghi et al., the authors should now present this paper by Luo et al. and explain honestly the differences. [complete ref : Luo, L., Liang, X., Ma, B., Zhou, H., 2021. A karst networks generation model based on the anisotropic Fast Marching algorithm. J. Hydrol. 600, 126507. https://doi.org/10.1016/j.jhydrol.2021.126507 ].*

Response:  We have added a reference to Luo et al., as requested. However, we would like to note that this reviewer comment is not fully correct: nowhere in the previously revised version of our manuscript do we claim that we were the first to employ anisotropic fast marching. We only explain that we test an assumption with a given model and we describe that model (pyKasso with anisotropic fast marching). We have inserted a brief mention of Luo's work in the text and briefly describe the major differences between the two approaches in line 151. An in-depth comparison of the methods is beyond the scope of this paper.

In addition, we would like to highlight three facts:

1. The primary concern of our study is not the development of anisotropic fast marching methods for simulating karst conduit networks. It is the application of these methods to hypothesis testing in a real catchment, regardless of the timing and development of these methods. In our previous round of revisions in response to reviewer comments, we removed several sections of text to focus more specifically on hypothesis testing rather than re-explaining method development.
2. The two approaches are quite different from one another, but we are certainly interested in comparing and perhaps merging the two approaches for future applications! A brief table summarizing the differences is provided below for the interest of the reviewers and editors, but will not be incorporated into the manuscript. We particularly note that, because Luo's approach does not allow for quickly generating multiple conduit network realizations of the same system, it would not be possible to use it for the type of hypothesis testing presented in our paper.
3. After examining the publication timelines, we found that the points raised in this comment are likely a result of publication delays. Our paper initially introducing our use of anisotropic fast marching for modeling karst networks was submitted on May 1st, 2021, one month *before* the paper by Luo et al. was published on June 1st, 2021. The paper by Luo et al. went through an unusually rapid review process (< 2 months from submission to publication). Our paper instead went through an unusually slow review process (10 months from submission to publication, despite the paper being accepted with only minor revisions). When we initially wrote about the use of anisotropic fast marching for karst network generation, it was indeed the first demonstrated use of this method, as the paper by Luo et al. had not yet been published. Additionally, several public presentations of our work were given between October 2020 and April 2021, introducing the idea of using anisotropic fast marching to simulate karst networks, all prior to Luo's publication. However, now that another research team has begun exploring similar methods, we are delighted to have more examples of this approach to draw on.

| Feature | Luo et al. | Fandel et al. |
|---|---|---|
| Conduit generation algorithm | Anisotropic Fast Marching | Anisotropic Fast Marching |
| Field data requirements | 1. Spatial extent of major hydrogeologic units
2. Locations of system inlets and outlets
3. Descriptive statistics for fracture network | 1. Spatial extent of major hydrogeologic units
2. Locations of system inlets and outlets
3. Descriptive statistics for fracture network |
| Number of dimensions | 3D | 2D (with 3D in development) |
| Discrete Fracture Network | Generated using MATLAB ADFNE1.5 toolbox (Algalandis, 2018) | Generated using built-in Python function in pyKasso, based on concepts in Baghbanan & Jing (2007) & Davy et al. (2013). |
| Travel cost | Not considered | Assigned based on hydro-geologic unit, and presence of fractures or existing conduits |
| Anisotropy field | Calculated based on hydraulic conductivity tensor of representative elementary volume voxel | Multiple options:
1. Orientation of geologic contacts between units
2. Land surface orientation
3. Hydraulic gradient orientation
4. Any other surface provided by the user |
| Stochasticity | Not considered | Multiple sources:
1. Discrete Fracture Network generation
2. Inlet/outlet locations & numbers
3. Inlet/outlet pairings |
| Influence of existing conduits | Not considered | Two options:
1. Known conduits can be included in the model from the beginning
2. Multiple iterations can be run for the same system, in which conduits generated by earlier iterations are considered in later iterations |
| Computational efficiency | Up to 56 hours for a single step (finding the equivalent hydraulic conductivity of a single REV subdomain in a single layer) | Under 2 minutes for 100 complete simulations (fracture network and conduits) |
| Language | MATLAB | Python |
| Availability | Closed-source (not public) | Open-source (source code and documentation of all components and libraries are publicly available) |

*Comment 3: As a consequence, the contribution should only focus on the idea to test 2 hypotheses of formation with a conduit network model ensemble, and put aside the "AFMA" part which was already the contribution of Fandel et al., 2022. Several proposals are made in that goal in the latter "On the flow" comments.*

*About the references, the authors should cite the founding works, and not just the papers of Fandel which re-implement existing methods:*

*• About AFMA (for example):*

*o Sethian (1999), implemented by Konukoglu et al. (2007); and Mirebeau et al. 2014. [Sethian, J.A., 1999. Level Set Methods and Fast Marching Methods Evolving Interfaces in Computational Geometry, Fluid Mechanics, Computer Vision, and Materials Science (second edition).*

*o Konukoglu, E., Sermesant, M., Clatz, O., Peyrat, J.M., Delingette, H., Ayache, N., 2007. A recursive anisotropic fast marching approach to reaction diffusion equation: application to tumor growth modeling. Inf. Process. Med. Imaging 4584, 687–699. https://doi.org/10.1007/978-3-540-73273-0_57*

*o Mirebeau, J.M., 2014. Anisotropic fast marching on Cartesian grids, using lattice basis reduction. SIAM J. Numer. Anal. 52 (4), 1573–1599. https://doi.org/10.1137/ 120861667.]*

*• For the modelling method:*

*o Cite Borghi et al. 2012.*

Response: We have inserted references to most of the key works mentioned above in Section 5 of the manuscript. Because the previous round of reviewer comments emphasized *removing* the sections of the text describing the development of the anisotropic fast marching techniques used in the current paper, we point readers to our previous work for a full discussion of the development of the methods and for grateful acknowledgement of the work by other authors upon which we built our approach.

*Comment 4: I still have reservations about the interpretations of the modelling results, but if the discussion is nuanced, it could be ok. In particular, the discussion should moderate the results of the model considering that:*

*o In pyKasso, and other similar approaches, the relative costs ("speeds" in AFMA) have been demonstrated to have an impact on the results. Here the costs are fixed (similar to those of Fandel 2022). Should other influencing factors have impacted the results? In particular, should other relative costs (less impact of the "distance to the source" relatively to "hydraulic gradient") have induced less attraction to QO in paths from N6 and N11 during glaciation?*

Response:

We have inserted a block of text in Section 7 that addresses both this comment and Comment #5 below. Essentially, the question is whether using a different parameter set (travel cost distribution, importance of the anisotropy field, etc.) for pyKasso could have resulted in a different conclusion.

In this comment, the reviewer states that the travel costs are fixed. However, in fact, we vary the travel costs both spatially and temporally. Different locations within the study area have different travel costs (limestone has a lower cost than insoluble units, fractures have a lower cost than limestone, etc.), and the travel costs in the scenario representing Hypothesis 1 are differently distributed than the travel costs in the scenario representing Hypothesis 2. In Hypothesis 1, during glaciation, the travel cost through areas

covered by the glacier is higher than in limestones not covered by the glacier, which does impact which paths conduits take, since the conduits will tend to avoid glaciated/high cost zones (see Figure 4 – some of the predicted paths under Hypothesis 2 would go through glaciated zones under Hypothesis 1, and indeed, these paths are not seen in the Hypothesis 1 scenario). Additionally, the fractures (not shown in Figure 4 because each simulation has a unique fracture network) also have a lower travel cost than the surrounding limestone, which influences the predicted conduit paths. In some cases, low-cost fractures lead to paths going more directly towards QO, and in other cases, the fractures direct paths away from QO. Because the manuscript already includes a discussion of travel cost distribution, we have focused our response more on addressing Comment #5, on the influence of the anisotropy field.

*Comment 5: In their answers as well as in the text, the authors often refer to "hydraulic gradient". I completely agree with them that it is what controls the flows in saturated zones. But in their simulations, the hydraulic gradient is approximated by "gradient of the lower contact between karstifiable unit and [the impermeable layer below]" (lines 180-185). Thus, it is probably constant over times, and could not allow an "inversion" from QA to QO as the hypothesis 1 would require, and as the authors described in some particular circumstances.*

*Said differently, the modelling approach gives a high importance to the "shortest path" between an inlet and an outlet. It is only slightly mitigated by fractures, and the AFMA is here to mitigate it by the "inception horizon" gradient (lines 180-185). As it is not directed QA to QO in the H1 test, reproducing the inactive conduit with the proposed approach would have been very surprising. => The results remain interesting, but nuance and moderation are required in the discussion and conclusion.*

Response: We have added text discussing this in Section 7. The gradient of the lower contact of the karstifiable unit is indeed constant in time in these simulations. However, the mapped portion of the inactive conduit network lies approximately along a line of equal elevation (Figure 1), which also roughly coincides with the edge of the glaciated extent as determined from geomorphological field evidence. It is therefore not at all immediately obvious to predict what path conduits might take in this zone, where the influence of the gradient and the influence of the travel cost overlap. It would have been entirely plausible for the conduits to follow the maximum downward gradient southeast until they encountered the glaciated high-cost zone, and then to follow along the contour (neither up nor down gradient) until they reached QO. Our initial expectation was in fact that the stochastic fracture distribution would slightly favor such a path in some simulations, while slightly favoring a more direct path ESE in other cases, resulting in a much broader spread of simulated conduit paths than what we actually saw.

*Comment 6: In the discussion and/or earlier in the text, it lacks "time lines" and explanations:*

*o Now the text is clear that the "glaciation phase" as well as the "covering of QS" have both happened. I suppose that the "covering" is after "glaciation", but it is not explicitly said: could you provide an approximative datation of these events or at least provide an relative chronology of events?*

Response: In line 140, we mention that the temporal order of events influences the conduit formation. We do not have enough clear field evidence to determine whether the glaciation preceded or followed the uncovering of QS. We have now stated more clearly in this paragraph that the chronology is uncertain.

*Comment 7: What has provoked the release of QS ? Sudden event or abrupt one?*

Response: Again, we do not have enough clear field evidence to answer this question.

*Comment 8: If H1 is not the explanation for the inactive Holloch conduit, but N6 + n11 existed and were drained by Qo, how do you imagine today's network (combining active + inactive) ? Should we need to "superpose" the conduits simulated by H1, to those by H2, and then to the active ones ?*

Response: This is discussed in the text in lines 275-281. It is indeed possible that there are conduits connecting N6 and N11 to QO, which have not yet been explored. The existence of such conduits and the existence of the mapped inactive conduits are not mutually exclusive. Since this is already discussed in the manuscript, no changes have been made to the text.

*Figure 1 : The cross-section line on the map is not consistent with the cross-section. I already mentioned indirectly this by asking to position the projection of N1, N6, N11 and N16 on it. Here is the fact : if you look at the map, along the cross-section starting from A, non-karst area (violet) is touching limestone (green) in a valley before quaternary sediments (yellow) on the north flank. On the cross-section, violet touches yellow which is in the valley… It should be corrected.*

Response: The cross-section is schematic to best show the general patterns of relationships between different units, and the map lumps all the different geologic units in the non-karst area together, such that the quaternary units are not shown. The cross-section also does not show the quaternary units outside of the main valley. The map has been updated to show more detail in the non-karst area, and the cross-section and caption have also been updated.

*Figures 3 and 4: why the isolines are those of the topography and not the isolines for the bottom of the limestones ? I hope it is just for the figures and the simulations consider the correct isolines (the ones of the bottom of limestones, as said in the new text)?*

Response: The contour lines on the figures show the land surface topography so that the reader can visualize the site. However, the surface used for simulation is the bottom of the limestone. The two surfaces are often roughly parallel, as can be seen in Figure 1c. We have updated the figure legends to clarify that the lines shown are topographic contours.

*Comment 9: The changes made in the model to be consistent with the author's own hypothesis (a development along the bottom of the limestones => thus a 2d model which considers the right geological formation at this level) imply quite different results clearly visible in figure 3, that, fortunately, do not modify the contribution nor the main conclusions of their previous work => I think it could be good to clearly assume that these new results are more consistent than the previous one and should be considered instead. Everyone could make an error, it would be clearer for the reader to assume as it is (and even say the "error" word in their text).*

Response: Thank you for pointing this out. We have added text in line 190 clarifying that the conduit networks simulated using the contact surface should be considered as more realistic representations of the system than those we previously published.

*On the flow comments (lines refer to the diff version):*

*Line 18 (abstract): suppress "(built on anisotropic fast marching methods)" because it is not the core of the paper (see comment1)*

Response: Because this paper does rely heavily on anisotropic fast marching methods, and because readers often rely on the abstract to identify papers of interest to read, we prefer to keep these keywords in the abstract so that readers interested in applications of anisotropic fast marching methods can more easily find this work.

*Lines 54-55 : rephrase (because the method has already been presented in previous papers, and to ease better understanding the contribution) : "This study USES a model-based approach to identifying the geological processes WHICH COULD EXPLAIN A PARTICULAR cave formation. IN this real karst SYTEM, detailed cave maps (...)"*

Response: We have revised this text based on these suggestions.

*Lines 96-100 : OK, but could you precise in the text if you consider this "expected network" as an exact location or an "approximative equivalent network". For what I understand, you have several clues on the "global path", but still some uncertainties about its precise location (as the differences between the mapped Holloch cave and the "expected network" at the same place demonstrate).*

Response: We have inserted text in line 97 to clarify that the expected network does not represent exact location, but rather the general configuration and approximate location of the conduits.

*Line 107 : remove Fandel et al. 2022 : no flow modelling is performed in this paper.*

Response: We have removed this paper from this list.

*Lines 185-190 about the "stochasticity": In Fandel et al.2022, it was said (legend of fig 18. A) "when the inlet/outlet are kept fixed, the fractures add only a small amount of variability in the network structure". The authors should remind this important aspect of their approach, because here, pairing is fixed. Thus, fracturation is the only remaining source of stochasticity.*

Response: We have inserted text in line 166 clarifying this point.

*Figure 3 : what are the paths on figure 3.c : they seem different from the "expected net" ? precise it in the legend*

Response: Figure 3c is intended to show the extent of the karstifiable limestone. One of many possible conduit network simulations is shown embedded in the limestone, but this is not the primary point of this panel. We have therefore removed the conduits from this panel.

*Line 260 : "in" is missing : "prediction are more certain IN some areas(...)"*

Response: Thank you for pointing this out. It has now been corrected in the manuscript.

*Line 300 : "better under hypothesis 1", I suppose you mean "hypothesis 2".*

Response: Thank you for pointing this out. It has now been corrected in the manuscript.

*Line 349 : "two-phase kastification" : if evidence of sink in glaciation times exist, don't we have three-phase kastification ?*

Response: Thank you for pointing this out. It has now been corrected in the manuscript.

---

## Author Response (AR3)

**Response letter for HESS-2022-280 Round 3**

**Title:** Improving understanding of groundwater flow in an alpine karst system by reconstructing its geologic history using conduit network model ensembles

**Authors:** Chloé Fandel, Ty Ferré, François Miville, Philippe Renard, Nico Goldscheider*

* corresponding author

**Editor's comment:** The manuscript has been reviewed by the most critical Reviewer involved in the prior round of reviews. While I recommended a set of major revisions to be implemented by the Authors, it is clear from the reviews that some additional clarifications are required at this stage. I am strongly recommending the Authors to consider the issues raised by the reviewer and incorporate their answers to these in their revised work.

**Response:** Thank you for this decision. We have made the requested changes, described in our detailed responses to reviewer comments below.

**Materials included in this submission**

1. The revised manuscript with markup showing the changes made
2. A point-by-point-response list
3. A clean version of the revised manuscript

**Point-by-point response to referee #2:**

*Comment 1: In this updated manuscript, the authors have made small changes. While some sentences have been added, I must admit I am somewhat disappointed. Despite some limitations I have identified, the current version is now more focused and allows future readers to grasp the overall work done. Moving forward, my remarks below, addressing the comments from the previous round, will be my final input. I trust the editor will consider them as they see fit.*

Response: We thank the reviewer for taking the time to provide thoughtful feedback over multiple rounds of review for this paper. We made additional corrections to ensure that the most recent version of the manuscript addresses the remaining comments, and is more satisfactory than the previous version.

*Comment 2: 1) yes, the author do not explicitly mention in the current manuscript that they were the first to employ anisotropic fast marching, BUT they acknowledged it as the second point in their online comments (visible online, in their first response of AC2, end of the paragraph…) and also in their response to the reviewers.*

*2) I already pointed out in my comments that the two approaches are indeed different, so it is disheartening to read an answer that implies I do not understand something that I have already myself explained to the authors. The same applies to the timeline. I was already aware of it.*

*3) They included a reference to Luo's work; however, they seem to downplay Luo's contribution in comparison to Fandel's. They emphasize that their own implementation is more general, but their arguments lack strength. The fact that Luo's paper does not explore the costs does not necessarily mean it is not feasible, as both works are based on Borghi's algorithm. Additionally, they overlooked mentioning that Luo's work is in 3D, while Fandel's current implementation remains in 2D. I want to clarify that I find Fandel's work interesting, but I am concerned about the authors' persistent attempts to diminish the significance of other researchers' contributions." => See the following comment, I propose a re-writing of lines 145-175.*

Response: We did not intend to downplay the importance of Luo et al.'s contribution – it simply is not as applicable for the purposes of this particular paper, in which the most important attributes are quick computation times and ability to consider multiple different influences (beyond fractures) on travel cost for conduit formation (e.g. rock type or surface cover). The ability to model the system in 3D is less important (although the 3D version of pyKasso is now functional as well). We have re-written lines 145-175 to clarify these points (see answer to comment below).

*Comment 3: The other references are also not well located to refer correctly to each one's contribution. More accurate citations would be (lines 145-175) (modified parts in capitals):*

*"To test these hypotheses, many possible network configurations were modelled using A STOCHASTIC SIMULATION METHOD, implemented in the Python karst modelling package pyKasso (Fandel et al., 2022). THIS PACKAGE IMPLEMENTS IN 2D, THE SKS APPROACH ORIGINALLY PROPOSED BY BORGHI ET AL. (2012), BUT USES AN ANISOTROPIC FAST MARCHING ALGORITHM (SETHIAN, 1999, MIREBEAU, 2014). SKS APPROACH CALCULATES the optimal path from one point to another through a medium, in which the ease of travel varies both spatially and directionally. Karst conduits can be simulated using this type of algorithm based on the assumption that a conduit represents the fastest path from an inlet (such as a doline or swallow hole) to an outlet (a spring). Luo et al. (2021) proposed also to use an anisotropic fast marching IN A 3D SKS IMPLEMENTATION, IN ORDER TO INCORPORATE THE EFFECT OF A FRACTURE NETWORK WITHOUT EXPLICITLY GENERATING A DISCRETE FRACTURE NETWORK. IN the pyKasso package, THE anisotropic fast marching ALLOWS TO RENDER THE EFFECT OF THE TOPOGRAPHY OF THE SURFACE ON WHICH THE KARST DEVELOPS, AS IT IS 2D. In pyKasso, the travel medium represents the geologic setting, in which some rock units are more soluble than others (i.e. easier for conduits to travel through). Conduits are also assumed to form preferentially in certain orientations: in the direction 155 of the maximum downward hydraulic gradient, and/or along the dip direction of bedding planes (Audra and Palmer, 2015; Dreybrodt et al., 2005; Palmer, 1991)."*

Response: We thank the reviewer for their suggestions and have revised significantly this section to clarify that we value Luo et al.'s contributions. We tried to explain more precisely the differences between the two implementations and our choice of pyKasso to address the questions posed in this specific paper. We have added a paragraph describing circumstances in which Luo et al.'s implementation may be a better choice.

*Comment 4: Once more, upon reviewing the response, I get the impression that the authors believe I did not comprehend the work. They turn around the point I raised: when I speak about "fixed costs" I speak about the fact that they are constant per media and per hypothesis. In their previous paper, they themselves demonstrated the impact of the relative cost variation (for example between the surrounding media and the fractures). Also, they state that their cost vary temporally: in the manuscript there are no mention of simulations in several steps with changing costs between the steps (unless the updating due to the iterative aspect of path after path generation when more than one source-sink couple is considered: in that case, the only cost which is updated (from what we can guess from previous papers) is the one of the cells crossed by a simulated conduit.*

*However, the added paragraph line 309-320 perfectly renders what I underlined, and I am OK with it.*

Response: The lack of comprehension was on our end, not on the reviewer. We had some difficulty understanding what was being asked, and we apologize for our misunderstanding.

Indeed, the travel cost in each medium (rock type, fractures, conduits, subglacial zones) is fixed. This is because we wished to keep all variables other than those being tested constant (and because we do not

expect things like the solubility of the limestone to change over time). The variables being tested are the locations of the active outlets, and the extent of glaciation. From this comment, it appears that the reviewer is satisfied with the explanation in lines 309-320, so we have not made any further changes.

*Addition: reading again part 5.2, I realize that for the description of both hypothesis, the authors use fuzzy terms like "high travel cost" (line 229): the exact value should be given into brackets, for precision.*

Response: Thank you for pointing this out. We have inserted a table listing the travel costs associated with each geologic setting.

*Comment 5: Yes, the mapped portion of inactive conduits lies approximately along a line of equal elevation BUT the algorithm computes the SHORTEST path between an inlet and an outlet, not all the possible paths. (REMINDER: FMA is determinist, not stochastic). In the absence of fractures (which are said to merely affect the results) the shortest path is mainly driven by the "topographic gradient" of the bottom surface. A gradient goes to 1 point not 2 points. It does not make a detour to stay at the same "level". Thus, it is not controlled by the elevation of the outlet, but more by the orientation of the above isolines (with the mitigation of the others elements included in the cost which, here, are only the fractures).*

*Anyway, I do not see in the added paragraph in section 7 (lines 309-320) any answer to that point (even the one of the authors with which I disagree): the added part answers quite well comment 4, not comment 5.*

Response: The reviewer is right: FMA is deterministic. But as shown and explained in the previous papers (Borghi et al. 2012, Fandel et al. 2021, Fandel et al. 2022) there are several level of stochasticity within SKS and pyKasso. These methods do not simply connect the points that are the closest. This is illustrated on figures 3 or 4, where many different geometries are shown. They are all compatible with the general principle and key assumptions. And for each one, the procedure is influenced differently by the interactions between the slope of the base topography, the presence of different fractures, and the order in which we activate the different springs as the reviewer clearly points out in the end of this comment.

We corrected the paragraph to remind the reader that stochasticity arises also because of the hierarchy, but we did not change the main content of this paragraph because we think that it already explains without ambiguity that results that our results depend from our assumptions and the way we implemented them. Therefore, they may not be fully correct and we discuss precisely that point.

*Comment 6: Now with the fact that the chronology is uncertain, I am less convinced. But well.*

*Comment 7: What has provoked the release of QS? Sudden event or abrupt one? If the authors have no answers, this point should be added as an interesting point to mitigate the results in the discussion, not just ignored as it is now.*

Response: We added a paragraph addressing these questions:

"One limitation of this study is that the order of events and the rate of change in the landscape is not precisely known. Field observations suggest a late glacial to postglacial age of the Sägebach spring and a relatively rapid exposure. The lack of precise dating limits our understanding of the system as a whole."

*Comment 8: [If H1 is not the explanation for the inactive Hölloch conduit, but N6 + n11 existed and were drained by Qo, how do you imagine today's network (combining active + inactive)? Should we need to "superpose" the conduits simulated by H1, to those by H2, and then to the active ones?]:*

*There is nothing in lines 275-281 answering really this question.*

Response: We are sorry, but we do not understand this question. The reviewer asks what would happen if the hypothesis H1 were not correct, "but N6 and N11 were drained to QO". But the hypothesis H1 *is* that N6 and N11 drain to QO. The last paragraph of the discussion does address a similar question, and states that it's possible that the actual conduit network looks like a superposition of H1 and H2. Testing additional scenarios would be very interesting but is beyond the scope of the present paper.

*Comment 9: Partly OK: the modifications (made) are not just the topography, but mainly the "formations" encountered at this elevation, and so the cost affected to the 2D grid on which the computation is performed. This is the main source of the different results that have been obtained in this revision round. It is not just a topographic gradient. For the reader's understanding, this should still be better explained.*

Response: In revising this section to address this comment, we realized that in fact we had superfluous text that repeated the explanation of differences between the original version and the current version twice. We have therefore removed the repeated text and clarified and consolidated this explanation into lines 220-230.

*Comment 10: the remarks concerning the paragraph about stochasticity. Indeed, the authors claim to clarify the point line 166, but they inserted a sentence line 170, which is not accurate and lost the reader. The concerned paragraph is between line 162-166, and I insisted on the fact that they should say clearly here that in the proposed manuscript the pairing of inlet and outlet is fixed, and thus, as, they were themselves saying in their previous works, HERE the amount of variability is quite reduced (said again differently, everything rest on the DFN generation which is only a quite limited source of variability). Although the term "fixed pairing" is mentioned at line 240, it appears deep within the manuscript's details and may not readily convey the implications concerning stochasticity to a more casual reader.*

Response: We have now dedicated a specific block of text in Section 5.1 explicitly to describing the differences between the models presented in our 2022 paper and those presented in the current manuscript, including mention of the inlet/outlet pairings, so that it is easier for x reader to identify all the differences together (see response to previous comment). We have not added any text in section 5, because this section describes how pyKasso works *in general* and not which settings and modeling choices were applied to this *particular* case. Those details are in section 5.1.

*Figure 1: So, the cross-section is still inconsistent (even with the map modifications, because the problems were not there), but the "schematic" adjective seems enough for the authors to justify it. As a geologist, I do not understand what forbid to draw a consistent cross-section as the map is quite simple. But well. The caption modification consists in precising that the cross-section is "with the structure generalized in the Schwarzwasser valley".*

Response: The original idea of our schematic cross-section was to represent the highly complex 3D topographical and geological environment (e.g. axial culminations, bent fold axes, etc.) in the form of a conceptual hydrogeological model showing all relevant and characteristic features in one profile, e.g., the prominent rockfall mass in the valley was projected onto the section line, although it is just east of the section line. However, it seems that this approach has caused more confusion than clarity. So instead we created a conventional and realistic cross-section, albeit still slightly simplified and generalized, e.g., patchy and thin Quaternary deposits on the slopes are not shown in cross-section. In the new version, we also differentiate between "overlying (Cretaceous) sandstone and marl" and the "flysch", although both units have similar hydrogeological properties (low permeability, predominant surface flow).

---

## Author Response (AR4)

Karlsruhe, 4 October 2023

Dear Editors,

Uploaded please find all required files. Thank you for your efforts.

Best regards, Nico Goldscheider